# Suppressing Pyroptosis Augments Post-Transplant Survival of Stem Cells and Cardiac Function Following Ischemic Injury

**DOI:** 10.3390/ijms22157946

**Published:** 2021-07-26

**Authors:** Chang Youn Lee, Seahyoung Lee, Seongtae Jeong, Jiyun Lee, Hyang-Hee Seo, Sunhye Shin, Jun-Hee Park, Byeong-Wook Song, Il-Kwon Kim, Jung-Won Choi, Sang Woo Kim, Gyoonhee Han, Soyeon Lim, Ki-Chul Hwang

**Affiliations:** 1Department of Integrated Omics for Biomedical Sciences, Yonsei University, Seoul 03722, Korea; cylee083@gmail.com (C.Y.L.); ssh5043@naver.com (S.S.); yrigmb@nate.com (J.-H.P.); gyoonhee@yonsei.ac.kr (G.H.); 2Institute for Bio-Medical Convergence, College of Medicine, Catholic Kwandong University, Gangneung-si 25601, Korea; sam1017@ish.ac.kr (S.L.); 91seongtae@gmail.com (S.J.); songbw@gmail.com (B.-W.S.); ilkwonkim@ish.ac.kr (I.-K.K.); jungwonjian@gmail.com (J.-W.C.); doctor7408@gmail.com (S.W.K.); 3Brain Korea 21 PLUS Project for Medical Science, Yonsei University, Seoul 03722, Korea; jylee12334@gmail.com (J.L.); seohyanghee@gmail.com (H.-H.S.)

**Keywords:** pyroptosis, M1 macrophage, stem cells, miR-762, I/R injury

## Abstract

The acute demise of stem cells following transplantation significantly compromises the efficacy of stem cell-based cell therapeutics for infarcted hearts. As the stem cells transplanted into the damaged heart are readily exposed to the hostile environment, it can be assumed that the acute death of the transplanted stem cells is also inflicted by the same environmental cues that caused massive death of the host cardiac cells. Pyroptosis, a highly inflammatory form of programmed cell death, has been added to the list of important cell death mechanisms in the damaged heart. However, unlike the well-established cell death mechanisms such as necrosis or apoptosis, the exact role and significance of pyroptosis in the acute death of transplanted stem cells have not been explored in depth. In the present study, we found that M1 macrophages mediate the pyroptosis in the ischemia/reperfusion (I/R) injured hearts and identified miRNA-762 as an important regulator of interleukin 1β production and subsequent pyroptosis. Delivery of exogenous miRNA-762 prior to transplantation significantly increased the post-transplant survival of stem cells and also significantly ameliorated cardiac fibrosis and heart functions following I/R injury. Our data strongly suggest that suppressing pyroptosis can be an effective adjuvant strategy to enhance the efficacy of stem cell-based therapeutics for diseased hearts.

## 1. Introduction

Since the first study reported the beneficial effect and clinical relevance of stem cell-based therapeutics for heart disease [1], basic science and clinical experiences have worked together to further optimize the stem cell-based therapeutics to treat the damaged myocardium [2,3,4]. Nevertheless, currently available data indicates that the clinical benefit of stem cell-based therapeutics has been marginal and inconsistent [5,6]. There could be a number of reasons that account for such disappointing outcomes, and the low post-transplant survival rate of stem cells could be one of them. To date, two mechanisms with which the transplanted stem cells exert beneficial effects have been proposed: the engraftment and subsequent trans-differentiation of the transplanted stem cells to replace damaged cells [7,8] and/or the paracrine stimulation of the survival and recovery of damaged cells [9]. Regardless of the ongoing debate on which of them is the major mechanism, the long-term survival of the transplanted stem cells would be the prerequisite to maximize the beneficial effect of stem cell-based therapeutics.

By the nature of the common cell-based therapeutic strategy under which cell transplantation is usually performed post-injury, the transplanted stem cells are readily exposed to the hostile environment of the damaged tissue. For example, the stem cells transplanted for cardiac disease will immediately face the same hostile environmental cues that caused massive death of the host cardiac cells such as hypoxia, ischemia, and/or oxidative stress [10,11,12]. In such a dire environment, cells are subjected to a life-or-death decision, and various cell death mechanisms including apoptosis, necrosis, and pyroptosis can be activated as a result [13,14]. Among those different cell death mechanisms, pyroptosis is the most recently recognized form of programmed cell death. First named by Brad T. Cookson in 2001, pyroptosis is characterized by typical cell lysis and inflammatory cytokine release [15]. Pyroptosis differs from apoptosis in that it does not involve the typical membrane blebbing of apoptosis but does induce cell lysis, swelling, and pore formation, which are missing in apoptosis. Furthermore, caspase-1 mainly mediates the disassembly of the cells in pyroptosis, while caspase-3 is the primary regulator of apoptosis [13,16]. Although pyroptosis seems to be developed as a host defense against microbial infections [17], it can be also initiated by non-infectious conditions such as myocardial infarction (MI) [18,19], suggesting that pyroptosis also can be a viable therapeutic target for cardiovascular diseases.

To date, both canonical and non-canonical pathways have been identified in pyroptosis [20]. In canonical activation, pyroptosis proceeds in several steps: (1) caspase-1 activation by NLRP3 inflammasome, (2) production of inflammatory cytokines, such as interleukin 1 beta (IL-1β) and IL-18, (3) cleavage and secretion of IL-1β and IL-18, leading to cell membrane rupture and the release of cytoplasmic cell contents, such as pro-inflammatory cytokines, endogenous ligands, alarmins, and other danger-associated molecular patterns, causing cell death [21,22,23]. Although the non-canonical pathway also eventually leads to cell lysis and proinflammatory cytokine release, caspase 11 (also known as caspase 4 as a human homolog) activation is essential for non-canonical NLRP3 inflammasome activation [24]. Active caspase 11 can cleave gasdermin D (GSDMD), in which the N-terminal domain of GSDMD forms membrane pores, leading to pyroptosis and subsequent NLRP3 inflammasome activation [25,26]. The cytokines released from damaged cells during pyroptosis can cause additional pyroptosis in the surrounding tissues and cells [27]. Therefore, the probability of pyroptosis having a negative impact on the initial survival of the transplanted stem cells will be higher than that of apoptosis which does not involve the release of inflammatory cytokines. In fact, pyroptosis is now categorized as programmed necrosis which refers to necroptosis, pyroptosis, ferroptosis, and mitochondrial permeability transition (MPT)-dependent necrosis. However, the exact role and significance of pyroptosis have not been explored in depth [28].

Thus, in the present study, we have examined the role of pyroptosis in ischemia/reperfusion (I/R) injured hearts and its impact on the survival of transplanted stem cells to evaluate its potential as a viable target for enhancing the long-term survival of transplanted stem cells and subsequent therapeutic efficacy in treating cardiovascular disease.

## 2. Results

### 2.1. Macrophages Are Recruited Following I/R Injury to Heart

I/R injury induces an acute condition called sterile inflammation [29], and IL-1β recruits macrophages during sterile inflammation [30]. As shown in Figure 1A, I/R injury caused severe left ventricle (LV) damage in 7d, and the number of F4/80 (a well-established macrophage marker) positive cells in the damaged hearts were apparently increased, particularly in the infarct zone. The characteristic pro-inflammatory response of myocardial reperfusion injury is known to be manifested between 6 and 24 h post-reperfusion [31]. Therefore, the number of F4/80-positive macrophages in the heart of 1d post-I/R injury was evaluated. The number of F4/80-positive cells significantly increased in the border and infarct zone compared to that of the remote zone (Figure 1B). These data indicated that the F4/80-positive macrophages were recruited to the site of I/R injury during the early pro-inflammatory response and remained for up to 7 days following I/R injury.

### 2.2. Pro-Pyroptotic Mediator IL-1β Increases in I/R Injured Heart

A differential gene expression (DEG) analysis to compare the gene expression of the sham group and I/R injury group was conducted using left ventricular (LV) tissues collected 1d after I/R injury. The results of the DEG analysis are presented as a volcano plot. The genes whose expression changed more than 2-folds (*p* < 0.05) are shown in green (down-regulated genes) or in red (up-regulated genes) (Appendix A). To investigate the effects of increased macrophage infiltration into the ischemic hearts, inflammation-related genes (*n* = 84) were selected based on a database analysis (QuickGO; GO:0006954). Our data indicated that two genes are involved in both the pyroptosis and inflammation process and showed a significant increase in the I/R group (QuickGO; GO:0070269). The expression of IL-1β and NLRC4 were significantly increased by approximately 6- and 3-fold, respectively, in hearts during early I/R injury. Additionally, we also checked the expression change of CASP4, which is known to play an important role in non-canonical NLRP3 inflammasome activation, and however, it was not significantly changed (Appendix A). Quantitative analysis indicated significantly higher IL-1β expression in the hearts of the I/R group than in the hearts of the sham group (Figure 1D). Increased IL-1β expression was further confirmed by immunohistochemistry staining, in which IL-1β expression was higher in I/R heart sections than in control heart sections (Figure 1E). These results show that IL-1β is significantly increased in hearts with I/R injury.

### 2.3. Pyroptosis-Related Gene and Protein Expression Increases in I/R Injured Heart

To verify whether the observed increase of macrophage recruitment and IL-1β production was indeed related to increased pyroptosis, the expressions of representative pyroptosis-related genes (caspase-1, IL-1β, IL-18, caspase-11, and GSDMD) in I/R injured heart (d1, d3, and d7) were examined by real-time PCR. The expression of caspase-1 and GSDMD significantly increased from d1, peaked at d3, and decreased at d7, yet still remained significantly increased compared to that of the sham group. On the other hand, the expression of caspase-11 only significantly increased 1d after the I/R injury and then decreased to the basal level (Figure 2A, left panel). The expression of both IL-1β and IL-18 significantly increased in I/R-injured hearts for up to 7d compared to a sham group (Figure 2A, right panel). Similarly, the protein expressions of caspase-1, caspase-11, GSDMD, and IL-1β in the infarcted areas of the 1d post-I/R-injured hearts were significantly increased compared to that of the sham group (Figure 2B,C). To further analyze the spatial distribution of pyroptosis-related factors in the infarct area and its vicinity, the expressions of mRNAs and proteins in the remote, border, and infarct zone of 1d post-I/R injured heart were examined. The mRNA expressions of caspase-1, caspase-11, and GSDMD significantly increased in the infarct zone compared to that in the remote zone, and the expression in the border zone was intermediate but still significantly increased compared to that of the remote zone, except for caspase-11 (Figure 2D). The protein expression showed a similar trend, and importantly, a cleaved, active form of IL-1β was apparently increased in the infarct zone (Figure 2E).

### 2.4. Pro-Inflammatory M1 Polarized Macrophage Increases Pyroptosis-Related Gene Expressions in Human Adipose-Derived Stem Cells

To investigate the role of macrophages as a mediator of pyroptosis during I/R injury, macrophages were classically and alternatively activated to induce M1 and M2 polarization using LPS/IFN-ɣ and IL-4, respectively [32]. LPS/IFN-ɣ-treated Raw264.7 cells showed an increase of M1 polarization marker NOS2 and CD86 [33], and IL-4 treatment increased the expression of ARG, an M2 polarization marker [34], indicating differential polarization was achieved as intended (Appendix A). Importantly, the mRNA expression of IL-1β significantly increased only in M1 polarized macrophages compared to that of untreated control cells, while there was no significant change of IL-1β expression in M2 polarized macrophages (Appendix A).

First, to evaluate the paracrine effect of M1 polarized macrophages on cardiac cells, cardiomyocytes were treated with 24 h conditioned media (CM) of M1 polarized macrophages for 24 h and were also cultured under hypoxic condition for 24 h to examine whether hypoxia itself could be an independent factor to induce pyroptosis. The M1 polarized macrophage CM significantly increased the expression of caspase-1, caspase-11, GSDMD, and IL-1β both at mRNA (Appendix A) and at protein level (Appendix A) compared to untreated control, but no significant effect of hypoxia was observed. Such M1 polarized macrophage CM-dependent increase of those pyroptosis-related factors was also observed in cardiac fibroblasts. Only the M1 polarized macrophage CM significantly increased both the mRNA (Appendix A) and protein (Appendix A) expression of caspase-1, caspase-11, GSDMD, and IL-1β, in cardiac fibroblasts. These data suggested that the recruited M1 polarized macrophages may trigger the induction of pyroptosis in surrounding cardiac cells.

Finally, M1 polarized macrophage CM again significantly increased the expression of caspase-1, caspase-4, GSDMD, and IL-1β, while hypoxia itself had no significant effect in human adipose-derived stem cells (hASCs) (Figure 3A). Similarly, M1 polarized macrophage CM increased the protein expressions of those pyroptosis-related factors in hASCs (Figure 3B,C), and such increase of IL-1β production was also reflected as an increased amount of IL-1β in hASC culture media (Appendix A). Immunocytochemical analysis also showed that the increase of caspase-1, GSDMD, and IL-1β in M1 polarized macrophage CM-treated hASCs, as well as caspase-4 which mediates IL-1β release (Figure 3D,E) [35]. These results altogether supported the speculation that M1 polarized macrophages may induce the pyroptosis of the transplanted stem cells in an IL-1β-mediated paracrine manner.

### 2.5. IL-1β Mediates the Induction of Pyroptosis in hASCs

To confirm the role of IL-1β as a mediator of the pyroptotic changes observed in M1 polarized macrophage CM treated hASCs, recombinant IL-1β was utilized. Recombinant IL-1β (10 ng/mL) significantly increased the expression of pyroptosis-related factors both at mRNA (Figure 4A) and protein level (Figure 4B). On the other hand, knockdown of IL-1β using IL-1β-specific siRNA readily suppressed the expression and secretion of IL-1β from M1 polarized macrophages in a dose-dependent manner (Appendix A). When hASCs were treated with the CM of IL-1β knocked down M1 polarized macrophages, the pyroptotic changes shown in Figure 3 were abolished (Figure 4A,B). These results strongly suggested that the IL-1β released from M1 polarized macrophages was a major paracrine factor that relayed the pyroptotic signal from M1 polarized macrophages to hASCs.

### 2.6. miRNA-762 Regulates IL-1β Production during I/R Injury

To find miRNAs targeting IL-1β, 4 different miRNA-target predicting databases were used, and 13 miRNAs were commonly indicated by the 4 databases as IL-1β targeting miRNAs (Figure 5A). Subsequent cross-checking with DEG analysis (with cutoff criteria of fold change > 2 and *p* < 0.05) resulted in the selection of miR-762 as a final candidate miRNA. To confirm whether miRNA-762 targets IL-1β, miRNA-762 was transfected into hASCs. Delivery of exogenous miRNA-762 significantly suppressed the increase of IL-1β expression induced by M1 macrophage-CM in hASCs (Figure 5B). DEG analysis also indicated that miRNA-762 expression was reduced by approximately 50% in ischemic hearts compared to that in control hearts (Figure 5C). In addition, we also examined whether miRNA-762 could regulate other pyroptosis-related markers such as caspase 1, GSDMD, and caspase 4 under the same condition of Figure 5B, and miRNA-762 reduced these marker expressions while anti-miRNA-762 did not affect the macrophage CM-induced protein expression (Appendix A). Next, whether miRNA-762 binds to the 3′UTR of IL-1β was confirmed by luciferase assays (Figure 5D). MiRNA-762 has a target site in the 3′ UTR of IL-1β, and the results indicated that miRNA-762 suppressed luciferase expression (Figure 5E). Together, these data show that miRNA-762 plays a critical role in inhibiting IL-1β expression in hASCs under pyroptotic stress.

### 2.7. miRNA-762 Augmentation Increases Post-Transplant Survival of hASC and Improves Cardiac Function Following I/R Injury

To verify the impact of miRNA-762 on the post-transplant survival of hASCs, miRNA-762-transfected hASCs (hASC^miR*−*762^) were transplanted to the I/R-injured heart. For the detection of viable transplanted cells, PKH26-labeling was done prior to the transplantation. The number of PKH26-positive cells in the hASC^miR*−*762^ group was significantly higher compared to that of the normal hASC group (Figure 6A,B). Furthermore, the results of Masson’s trichrome staining of the heart collected 7d post-I/R injury indicated that the transplantation of hASC^miR*−*762^ well maintained LV wall thickness and significantly attenuated cardiac fibrosis (Figure 6C) as evidenced by the significantly decreased fibrotic area (Figure 6D). Transplantation of hASC^miR*−*762^ also significantly attenuated I/R-induced decline of heart function evaluated by stroke volume (Figure 6F), ejection fraction (Figure 6G), and stroke work (Figure 6H). The heart rate in each group was not significantly changed (Figure 6E). These data altogether suggested that the augmentation of miRNA-762 significantly promoted the post-transplant survival of hASCs and suppressed cardiac fibrosis, improving heart function.

## 3. Discussion

Among many different types of cardiovascular diseases, myocardial infarction (MI) may be the most closely related to pyroptosis since it involves massive cardiac cell death and inflammation. In the present study, we have demonstrated that I/R-injury to the heart causes M1 polarized macrophage-mediated pyroptosis that negatively affects the post-transplant survival of stem cells in an IL-1β-dependent manner. Furthermore, the present study shows that preventing pyroptosis of the transplanted stem cells by down-regulating the production of IL-1β could improve the post-transplant survival of stem cells and heart function.

Stem cells applicable to regenerative cell therapy include embryonic stem cells (ESCs), mesenchymal stem cells (MSCs), hematopoietic stem cells (HSCs), tissue/organ-specific stem cells [36], and induced pluripotent stem cells (iPSCs) [37,38]. For the present study, ASC, a type of adult mesenchymal stem cell, was used due to its easy isolation protocol, multi-lineage cell differentiation capacity, and regenerative potential [39]. Furthermore, it is relatively free from the ethical considerations and safety issues such as tumorigenicity and immunogenic response that impede the clinical relevance of ESCs or iPSCs [36,40,41]. However, it still cannot escape the issue of low post-transplant survival, which critically compromises the therapeutic efficacy of the transplanted stem cells [42,43]. To enhance the survival of transplanted stem cells, a number of approaches had been tried such as amplification of anti-apoptotic signals [44,45,46,47], fortification of initial cell adhesion [48,49,50], hypoxic preconditioning [51,52,53], and various soluble factors including growth factors, cytokines, and chemicals [54,55,56,57]. Nevertheless, still no single gold standard to guarantee the post-transplant survival of stem cells is currently available. Therefore, the need for alternative options to enhance the post-transplant survival of stem cells is well justified.

Pyroptosis, as a highly regulated cell death process, has been reported to occur in a wide range of cardiovascular diseases including, but not limited to, atherosclerosis, MI, hypertension, I/R injury, cardiomyopathy, and heart failure [58]. Although its potential as a therapeutic target for cardiovascular disease has been empirically explored [59,60], its exact role and significance in the post-transplant death of stem cells have not been explored in-depth, making this study possibly one of the first reports to demonstrate the critical role of pyroptosis in the acute demise of stem cells following transplantation. Previous studies have demonstrated that MSCs can alleviate inflammation including pyroptosis by secreting a variety of anti-inflammatory factors such as prostaglandin E2, interleukin-10, and interleukin-12 [61,62]. However, the survival rate of transplanted stem cells in the hostile environment of infarcted heart has been reported as around 1–6% at 60 min–24 h after injection [63,64], making it difficult for the residual transplanted stem cells to exert sufficient anti-inflammatory function. This may explain the moderate improvement of cardiac function in clinical trials using stem cell-based cell therapies.

In this study, we presented pyroptosis as one of the causes of poor survival of transplanted stem cells and utilized miRNA as a tool for proving the concept that miRNA-mediated down-regulation of IL-1β expression and secretion can prevent the pyroptosis of the transplanted hASCs and subsequently improve cardiac function. As the M1 polarized macrophages showed an increased IL-1β mRNA expression, it was postulated that M1 polarized macrophages recruited to the site of I/R injury propagate pyroptosis in a paracrine manner via secreting IL-1β, thus affecting host cardiac cells and stem cells transplanted to the site of injury. One thing that needs to be mentioned regarding this statement is that it is one of the limitations of the present study that we were not able to clearly demonstrate that miRNA-762 directly targets the 3′ UTR of IL-1β by using anti-miRNA-762 in luciferase assays. Although miRNA-762 significantly suppressed luciferase activity indicating miRNA-762 can prevent translation of IL-1β in the present study, the neutralizing experiment should have been conducted to definitely demonstrate direct targeting of IL-1β 3′ UTR by miRNA-762, and this issue has to be properly addressed in further studies. Another limitation also needs to be mentioned is that we used CM of mouse origin macrophage to stimulate hASCs. BLAST search indicated that mouse IL-1β is 78.65% identical to that of humans at mRNA (NM_008361.4) level and 68% identical at protein level (NP_032387.1). Although our various data collectively suggest that even IL-1β of mouse origin could stimulate human ASCs so that using CM of mouse origin for this proof of concept study is acceptable, it may be necessary for further studies to use IL-1β of human origin for the study to have clinical relevance.

One of the interesting findings of the present study is the miRNA-mediated regulation of pyroptosis in the I/R-injured heart. Over the last few decades, non-coding microRNAs (miRNAs) have emerged as important regulators of both physiological and pathological processes of virtually every disease, including cardiovascular disease [65,66,67]. To be more specific, IL-1β-targeting miRNA-762 decreased following I/R injury, causing the increased production of IL-1β that enabled pyroptotic signals to further propagate to surrounding cells including transplanted stem cells. Regarding the role of miRNA-762 in I/R injured heart, a very recent study used a mouse model that reported that I/R injury-induced translocation of miRNA-762 to mitochondria where it caused cardiomyocyte apoptosis by down-regulating NADH dehydrogenases subunit 2 (ND2) [68]. At glance, it seems that that particular study directly contradicts what we have demonstrated in the present study is mainly 2 aspects; the expression pattern of miRNA-762 after I/R injury is opposite and opposite modulation of miRNA-762 (miRNA-762 neutralization vs. exogenous miRNA-762 supplementation) produced similar functional improvement of the heart. Nevertheless, time-dependent changes of miRNA may be the key to explain such discrepancy.

Previous studies have demonstrated that the expression of certain miRNAs shows dramatic temporal changes as pathologic condition progresses [69,70,71]. Furthermore, the expression of a certain miRNA can be also bidirectional depending on the intensity of the same stimulation. In fact, our group has previously demonstrated that the hydrogen peroxide-induced expression of miRNA-1 in cardiomyocytes can be bidirectional depending on the concentration of hydrogen peroxide used [72]. According to the above-mentioned study reported I/R injury-induced increase of miRNA-762, the authors observed increased miRNA-762 expression for the first 60 min after I/R injury, and miRNA-762 expression was not further examined thereafter. On the other hand, we observed a significant decrease of miRNA-762 one day after I/R injury. Therefore, if miRNA-762 is one of the miRNAs whose expression pattern changes significantly with time following I/R injury so that the expression of miRNA-762 rapidly increased immediately after I/R injury and then decreased thereafter, the issue of the opposite expression pattern of miRNA-762 after I/R injury can be solved without mutually contradicting each study.

Furthermore, assuming the temporal expression change of miRNA-762 following I/R has occurred, suppression of the increase of miRNA-762 during acute phase using anti-miR can improve the outcomes following I/R injury by preventing initial cardiomyocyte death as their data demonstrated, while augmentation of the decreased miRNA-762 during latent phase can improve the outcomes following I/R injury by preventing pyroptosis of transplanted stem cells so that enabling them to exert a beneficial effect on surviving, remain cardiac cells as we demonstrated in the present study. Additionally, if the down-regulation of miRNA-762 in the present study is indeed induced by the initial massive death of cardiomyocytes following I/R injury as pyroptosis is, suppression of the increase of miRNA-762 during the acute phase may also prevent the down-regulation of miRNA-762 during the latent phase because the suppression of initial surge of miRNA-762 can prevent the initial massive death of cardiomyocytes, which in turn, triggers the process of pyroptosis along with the down-regulation of miRNA-762. However, these possibilities will remain speculative until they are empirically verified. To validate or rule out these possibilities, further study especially focused on time-dependent miRNA-762 expression change and elucidation of the underlying mechanisms is definitely required.

## 4. Materials and Methods

### 4.1. Culture of Human Adipose Derived Stem Cells

Human adipose-derived stem cells (hASCs) were purchased from Invitrogen (Waltham, MA, USA). hASCs were cultured according to the manufacturer’s instructions. hASCs were grown in low glucose Dulbecco’s modified Eagle’s medium (DMEM; Gibco, Waltham, MA, USA) supplemented with 10% heat-inactivated fetal bovine serum (FBS; Gibco) and 1% penicillin-streptomycin (Gibco). The medium was replaced with fresh medium every two to three days, and cells were passaged using 0.25% trypsin-EDTA (Gibco) when they reached about 80% confluency. Cells from passages 6 to 8 were used for experiments.

### 4.2. Raw264.7 Cells Culture and Culture Medium Preparation

The murine macrophage cell line Raw264.7 (Korean Cell Line Bank, Seoul, Korea) was cultured in DMEM (Gibco) supplemented with 10% FBS (without heat-inactivation, Gibco) and 25 mM HEPES buffer (Gibco) with 1% penicillin-Streptomycin (Gibco). Cells were cultured in 5% CO_2_/95% humidified air at 37 °C. The medium was changed every other day. For collecting macrophage-conditioned media (MØCM), the cells were cultured in a medium containing LPS (100 ng/mL; R&D Systems, Inc., Minneapolis, MN, USA) and IFN-γ (20 ng/mL; R&D Systems, Inc.) for 24 h. After the stimulation with LPS and IFN-γ, the medium was changed to new serum-free-DMEM and incubated for an additional 24 h. After the incubation, the macrophage-cultured medium was collected and centrifuged at 2000× *g* for 10 min. The supernatant was sterilized by filtrating through a 0.22 μm filter (Merck Millipore, Burlington, MA, USA), which is used as MØCM. The collected MØCM were kept at −80 °C until used.

### 4.3. Experimental Induction of Hypoxia

To induce hypoxia, hASCs were plated in cell culture dishes at 80% confluency. The serum-free low glucose DMEM was degassed. The hASCs were cultured in a hypoxic chamber (Thermo Fisher Scientific, Waltham, MA, USA) maintained below 1% O_2_ concentration at 37 °C for 24 h after 2 times washes with the degassed DMEM.

### 4.4. MiRNA Transfection

hASCs were plated in cell culture dishes at 80% confluency one day before transfection. Transfection with miRNA-762 mimics or miRNA-762 inhibitors was performed by the TransIT-X2 system (Mirus Bio, Madison, WI, USA). Transfection complex containing mature miRNA-762 mimics (Genolution Pharmaceuticals, Seoul, Korea) or miRNA-762 inhibitors (Integrated DNA Technologies, Coralville, IA, USA) were added directly to cells in a complete culture medium containing 10% FBS and no antibiotics at final concentrations of 20 nM. After the cells were incubated for 24 h, MØCM was treated for additional experiments.

### 4.5. Transient Knockdown of IL-1β

Transfection of siRNA was performed using the TransIT-X2 system (Mirus Bio). Commercial AccuTarget siRNAs, (Bioneer, Daejeon, Korea) which are mouse IL-1β siRNA (sense 5′-CAGGCUCCGAGAUGAACAA (dTdT)-3′; antisense 5′-UUGUUCAUCUCGGAGCCUG (dTdT)-3′), were designed to knockdown mouse IL-1β gene expression. Raw264.7 were plated in cell culture dishes at 80% confluency one day before transfection. IL-1β siRNA directly cells in complete culture medium containing 10% FBS and no antibiotics at final concentrations of 50 nM. To confirm the efficiency of IL-1β gene knockdown, immunoblot analysis and IL-1β ELISA using cell lysate and media, respectively, were conducted.

### 4.6. Reverse Transcription Polymerase Chain Reaction (RT-PCR)

Total RNA was extracted using TRIzol (Ambion, Waltham, MA, USA) according to the manufacturer’s instructions. Briefly, Chloroform (Sigma Aldrich, St. Louis, MO, USA) was added to the extract and was vortexed for 15 s. Next, centrifugation at 12,000× *g* at 4 °C for 15 min, caused three layers to appear, and the transparent upper layer was collected into a new tube. Each sample next received 2-propanol (Sigma Aldrich), and gently mixed for 10 s. Centrifugation at 12,000× *g* at 4 °C for 10 min. The supernatant was discarded and the pellet washed with 75% (*v*/*v*) ethanol admixed with 1% diethylpyrocarbonate (DEPC; Sigma Aldrich) contained water. Centrifugation at about 12,000× *g* at 4 °C for 5 min, followed. The supernatant was again discarded and the pellet dried at room temperature for one to two minutes. Finally, each pellet was dissolved in 30 μL nuclease-free water (NFW). The total RNA quality and concentration were measured using Nano-Drop Lite (Thermo Fisher Scientific). Complementary DNA (cDNA) was synthesized using a Maxime RT Premix kit (iNtRON Biotechnology, Seongnam, Korea) by the manufacturer’s methods. Briefly, one microgram of RNA was added to NFW. Each sample was incubated at 42 °C for 1 h then incubated at 95 °C for 5 min. One hundred nanograms of cDNA, 10 pM of each primer (forward and backward; Table 1), and EmeraldAmp GT PCR Master Mix (Takara Bio Inc., Shiga, Japan) were mixed with NFW to give a final volume of 25 μL. PCR conditions were as follows. A cycle of denaturation at 95 °C for 3 min was followed by 25~30 elongation cycles each featuring denaturation at 95 °C for 15 s, annealing at 48 to 60 °C for 15 s, and elongation at 72 °C for 15 s. The reaction was next held at 72 °C for 10 min. PCR products were separated by electrophoresis in 1.2% (*w*/*v*) agarose gels (BioRad, Hercules, CA, USA), and Gel-Doc (Bio-Rad, Hercules, CA, USA) was used to visualize bands after staining with RedSafe ^TM^ (Intron Biotechnology, Seongnam-si, Korea).

### 4.7. Real-Time Quantitative Polymerase Chain Reaction (qRT-PCR)

The cDNA was obtained in the same way as above. Amplification and detection of specific products were performed on a StepOnePlus Real-time PCR System (Applied Biosystems, Waltham, MA, USA) using SYBR Premix Ex Taq II (Takara Bio Inc.) at 95 °C for 30 s, followed by 40 cycles of 95 °C for 5 s and 60 °C for 30 s. The threshold cycle (Ct) of each target gene was automatically defined and normalized to control GAPDH (ΔCt value). The relative difference in expression levels of each mRNA in hASCs (^ΔΔ^Ct) was calculated and presented as fold induction (2^−ΔΔ^Ct). Table 1 describes the information about primers used in this work.

**Table 1 ijms-22-07946-t001:** PCR primers used in this study.

Gene	Sequence (F)	Sequence (R)
Mouse IL-1β	5′–GGAGAACCAAGCAACGACAAAATA–3′	5′–TGGGGAACTCTGCAGACTCAAAC–3′
Mouse ARG1	5′–ATGGAAGAGACCTTCAGCTA–3′	5′–GCTGTCTTCCCAAGAGTTGG–3′
Mouse NOS2	5′–TGCTGTTCTCAGCCAACAA–3′	5′–GAACTCAATGGCATGAGGCA–3′
Mouse CD86	5′–TCCTGTAGACGTGTTCCAGA–3′	5′–TGCTTAGACGTGCAGGTCAA–3′
Mouse GAPDH	5′–GGGTGTGAACCACGAGAAATA–3′	5′–GTCATGAGCCCTTCCACAAT–3′
Rat Caspase1	5′–GGGCAAAGAGGAAGCAATTTATC–3′	5′–GCCAGGTAGCAGTCTTCATTAC–3′
Rat Caspase11	5′–GTGTGGATCAGAGAGTCTTCAG–3′	5′–GGTGTGGTGTTGTAGAGTAGAG–3′
Rat GSDMD	5′–TTTAGTCTGCTTGCCGTACTC–3′	5′–GCTGTTGTCTCAACTCCATTTC–3′
Rat IL-1β	5′–GAGGACATGAGCACCTTCTTT–3′	5′–GCCTGTAGTGCAGTTGTCTAA–3′
Rat IL-18	5′–AATGGAGACTTGGAATCAGACC–3′	5′–GCCAGTCCTCTTACTTCACTATC–3′
Rat GAPDH	5′–ATGGAGAAGGCTGGGGCTCACCT–3′	5′–AGCCCTTCCACGATGCCAAAGTTGT–3′
Human Caspase1	5′–GGAAACAAAAGTCGGCAGAG–3′	5′–ACGCTGTACCCCAGATTTTG–3′
Human Caspase4	5′–AAGAGAAGCAACGTATGGCAGGAC–3′	5′–GGACAAAGCTTGAGGGCATCTGTA–3′
Human GSDMD	5′–AGTGTGTCAACCTGTCTATCAAG–3′	5′–ACACTCAGCGAGTACACATTC–3′
Human IL-1β	5′–TGAGCTCGCCAGTGAAATGA–3′	5′–AGATTCGTAGCTGGATGCCG–3′
Human GAPDH	5′–CATGGGTGTGAACCATGAGA–3′	5′–GGTCATGAGTCCTTCCACGA–3′

### 4.8. MicroRNA Quantification

Total RNA was extracted using TRIzol (Ambion) according to the manufacturer’s instructions. Then, 100 ng purified total RNA was used for reverse transcription (TaqMan^®^ MicroRNA Reverse Transcriptase Kit, Applied Biosystems) in combination with TaqMan MicroRNA Assays to quantify miRNAs and U6 control transcripts according to the manufacturer’s conditions. The threshold cycle (C_t_) of miRNA-762 and U6 expression was automatically defined, located in the linear amplification phase of the PCR, and normalized to the control U6 (ΔC_t_ value). The relative difference in the expression level of miRNAs in the sorted cells (ΔΔC_t_) was calculated and presented as the fold induction (2^−ΔΔCt^).

### 4.9. Immunoblot Analysis

To obtain proteins, freshly isolated hearts were cut into small pieces using surgical scissors and then placed in a ReadyPrep Mini grinder (BioRad). The tissue was lysed by RIPA buffer (Thermo Fisher Scientific) containing 1% phosphatase inhibitors (Roche, Basel, Switzerland) and 1% protease inhibitors (Roche). hASCs were washed with PBS and then lysed by RIPA buffer (Thermo Fisher Scientific) containing 1% phosphatase inhibitors (Roche, Basel, Switzerland) and 1% protease inhibitors (Roche). Protein concentrations were determined by the BCA Protein Assay kit (Thermo Fisher Scientific). 15 µg protein was loaded to 10% or 12% sodium dodecyl sulfate-polyacrylamide gel electrophoresis (SDS-PAGE) and then transferred to a polyvinylidene difluoride membrane (PVDF, Millipore, Billerica, MA, USA). The membrane was blocked with 5% skim milk diluted in tris-buffered saline with 0.05% tween 20 (TBS-T for one hour at room temperature and then incubated with the primary antibodies overnight at 4 °C. The following antibodies and dilution ratio were used in these experiments: anti-caspase-1 (1: 1000, ab1872, Abcam), anti-caspase-4 (1:500, sc-56056, Santa Cruz Biotechnology, Dallas, TX, USA), anti-caspase-11 (1:1000, sc-374615, Santa Cruz Biotechnology), anti-GSDMD (1:1000, sc-393581, Santa Cruz Biotechnology), anti-IL-1β (1:1000, NB600-633, Novusbio), and anti-β-actin (1:2000, A5316, Sigma Aldrich). The membrane was washed with 0.01% TBS-T for five minutes three times and then incubated with 5% skim milk and horseradish peroxidase-conjugated secondary antibodies (Santa Cruz Biotechnology) for one hour at room temperature. After the membrane was washed six times for five minutes, the bands were detected with an enhanced chemiluminescence (ECL) reagent (Abclon Inc., Seoul, Korea). The band intensities were quantified using a Davinch-Western imagine system (Davinch K, Seoul, Korea) and NIH ImageJ version 1.44p software (National Institutes of Health, New York, NY, USA).

### 4.10. Immunocytochemistry (ICC)

hASCs were plated in 4-well glass chamber slides (SPL Life Science, Pocheon-Si, Korea) and cultured. After treatment, the cells were washed three times using PBS and fixed in 4% paraformaldehyde for 10 min. The cells were washed again with PBS, followed by blocking with 0.5% bovine serum albumin (BSA) at room temperature and incubation with anti-caspase-1 (1:200, ab1872, Abcam), anti-caspase-4 (1:100, sc-56056, Santa Cruz Biotechnology), anti-GSDMD (1:100, sc-393581, Santa Cruz Biotechnology), anti-IL-1β (1:100, NB600-633, Novusbio) antibody overnight at 4 °C. After washing three times with PBS, the cells were incubated with FITC-conjugated goat anti-rabbit IgG and rhodamine-conjugated rabbit anti-mouse IgG at room temperature for 1 h in the dark. The cells were mounted using DAPI-containing mounting medium (Santa Cruz Biotechnology) and then signals were examined using an Olympus BX53TR microscope (Tokyo, Japan). All images were processed using the CellSens software (Olympus).

### 4.11. Luciferase Assay

The UTR sequences of human IL-1β, including the XhoI (forward) and XbaI (reverse) endonuclease sites at both ends, were purchased from Integrated DNA Technologies (IDT, Republic of Singapore). The 3′ UTR fragment was then cloned into the pMIR GLO vector (Promega Corporation, Fitchburg, WI, USA). HeLa cells were plated at a density of 1 × 10^5^ cells/well in a 12-well plate and then transfected with either pMIR GLO control vector or pMIR GLO vector with IL-1β 3′ UTR using Lipofectamine 2000 (Thermo Fisher Scientific). Two days after the transfection, the luciferase activity of the cells was measured using a luminometer and Dual Luciferase Assay Kit (Promega Corporation) according to the manufacturer’s instructions. Cell number and transfection efficiency were normalized with Renilla luciferase assay. Luciferase intensity was determined using the Glomax^®^ Explorer multimode microplate reader (Promega, WI, USA).

### 4.12. Induction of I/R Injury and Cell Transplantation

All experimental procedures for animal studies were approved by the Committee for the Care and Use of Laboratory Animals of Catholic Kwandong University College of Medicine (CKU01-2017-002) and performed in accordance with the Committee’s Guidelines and Regulations for Animal Care. The ischemic myocardial infarction rat model was established by using a method described previously [49]. Seven -weeks-old Sprague-Dawley male rats (220 ± 30 g) were assigned into five groups (Sham, I/R control, I/R with ASC, I/R with ASC + miRNA-762, and I/R with ASC + miRNA-762 inhibitor). Briefly, under general anesthesia with Zoletil^®^50 (0.8 mL/kg; Virbac, Frances) and Rompun (0.2 mL/kg; Bayer Korea CO, Korea), rats were intubated, and positive-pressure ventilation (12 mL/kg) was maintained with room air supplemented with oxygen (70 strokes/min, tidal volume: 8–10 mL/kg) using Harvard ventilator (Harvard Apparatus, Holliston, MA, USA). Then, the third and fourth ribs were cut to open the chest, and the heart was exteriorized through the intercostal space. The heart was exposed through a 2-cm left lateral thoracotomy. The pericardium was incised and a 7-0 silk suture (Johnson & Johnson, Langhorne, PA, USA) was placed around the proximal portion of the left coronary artery, beneath the left atrial appendage. Ligature ends were passed through a small length of plastic tube to form a snare. For coronary artery occlusion, the snare was pressed onto the surface of the heart directly above the coronary artery, and a hemostat was applied to the snare. Ischemia was confirmed by the blanching of the myocardium and dyskinesis of the ischemic region. After 60 min of occlusion, the hemostat was removed and the snare released for reperfusion, with the ligature left loose on the surface of the heart. Restoration of normal rubor indicated successful reperfusion. For cell transplantation, PKH26-labelled cells were suspended in 30 µL of PBS (1 × 10^6^ cells) and transplanted into the viable myocardium bordering the infarction at three injection sites using an insulin syringe (BD Ultra-Fine II, 0.3 mL) with a 30-gauge needle. Wounds were sutured and the thorax was closed under negative pressure. Rats were weaned from mechanical ventilation and returned to cages to recover. In sham-operated rats, the same procedure was executed without tightening the snare. Depending on experiments, the hearts were collected on 1 day to 3 weeks after I/R.

### 4.13. Total RNA Sequencing and Data Analyses

Total RNA was extracted using TRIzol reagent (Ambion) according to the manufacturer’s instructions from freshly isolated hearts. RNA quality was estimated by an Agilent 2100 bioanalyzer using an RNA 6000 Nano Chip (Agilent Technologies, Amstelveen, The Netherlands) and RNA quantity was determined by an ND-2000 Spectrophotometer (Thermo Fisher Scientific, Wilmington, DE, USA). Library production was performed by a SMARTer Stranded RNA-Seq Kit (TaKaRa Bio, Mountain View, CA, USA) and High-throughput paired-end 100 sequencing was done by HiSeq 2500 system (Illumina, San Diego, CA, USA). Total RNA sequencing reads were mapped using TopHat software [73] to obtain the alignment file. ExDEGA (e-Biogen, Seoul, Korea) was used for data mining and graphic visualization with criteria of *p* < 0.05, a fold change of 2, and a log2 normalized read count of ≥4. Further, gene ontology (GO) analysis was performed by DAVID (http://david.abcc.ncifcrf.gov/summary.jsp accessed on 23 May 2021) and included inflammatory response and pyroptosis signaling in functional gene classification.

### 4.14. Immunohistochemistry (IHC)

Hearts were excised 1 day or 1 week after I/R. The heart was perfused with Krebs-Ringer Bicarbonate Buffer (KRBB: 120 mM NaCl, 25 mM NaHCO_3_, 5 mM KCl, 1.2 mM KH_2_PO_4_, 1.2 mM MgSO_4_, 2.5 mM CaCl_2_, and 20 mM MOPS) for 10 min to wash out the blood and then fixed with 10% (*v*/*v*) neutral-buffered formaldehyde, overnight at 4 °C.

For IHC using cryosection, after washing with PBS three times, the tissues were incubated with 30% sucrose in PBS overnight. Then tissues were embedded in OCT compound 4583 (Tissue-Tek; Miles Inc., Elkhart, IN, USA) and stored at −80 °C until needed. Cryosections of 5 μm of thickness were obtained using HM525 NX Cryostat (Thermo Fisher Scientific) and mounted on gelatin-coated glass slides to ensure that different stains could be used on successive sections of tissue cut through the areas of I/R injury. After tissue sections were rehydrated in PBS for 5 min, then treated with PBS containing 1% SDS for 4 min. To remove flavin coenzyme autofluorescence, tissue sections were treated with 0.1% sodium borohydride for 30 min. After blocking with 1% bovine serum albumin (BSA) in PBS for 20 min, slides were incubated with primary antibodies at 4 °C overnight. FITC-conjugated goat anti-rabbit IgG and rhodamine-conjugated rabbit anti-mouse IgG were used as secondary antibodies. The sections with DAPI staining for nuclei were performed using Vectashiled^®^ antifade mounting medium with DAPI (Vector Laboratories, Burlingame, CA, USA) and were analyzed by confocal laser scanning microscope (LSM 700, Zeiss) and transferred to a computer equipped with ZEN2012 software. The areas are expressed as percentages of the total left ventricle. Anti-F4/80 (sc-377009, Santa Cruz Biotechnology) was incubated at a concentration of 1/100 on slides. For quantification of F4/80 and DAPI, five frozen hearts of each group were prepared and 3 fields from three different regions (remote, border, infarct zone) per one heart slide were chosen for observation. For detection of IL-1β, anti-IL-1β (NB600-633, Novusbio) was incubated at a concentration of 1/100 on sections overnight at 4 °C and was incubated with an HRP-labeled secondary antibody (Dako, Carpinteria, CA, USA). And sections were further processed using the DAB reagent kit (Dako) for visualization. Slides were examined using an Olympus BX53TR microscope (Tokyo, Japan). All images were processed using the CellSens software (Olympus).

For IHC using paraffin section, after washing with running tab water for 30–40 min, the tissues were dehydrated using Ethanol and then embedded in paraffin. 5 μm paraffin sections were obtained using a microtome, which was then deparaffinized in xylene and rehydrated in graded Ethanol and PBS. Antigen retrieval was done by heating in citrate buffer (pH 6.0) for 40 min. After cooling at 4 °C for 10 min, sections were rinsed with PBS and then blocked with 1% BSA in PBS. A detailed method of primary antibodies incubation and detection is the same as cryosection.

### 4.15. Hematoxylin-Eosin (H&E) Staining

The tissue was fixed in 4% formaldehyde and embedded in paraffin. Tissue sections (5 μm thickness) were deparaffinized, dehydrated, and rinsed with PBS. The tissue was stained by Hematoxylin-Eosin (H&E) stain kit (Sigma-Aldrich), following the manufacturer’s protocol. Briefly, after incubation in Hematoxylin Solution for 5 min, the slides were washed and stained with Eosin for 5 min and dehydrated in ethanol and xylene. H&E stained slides were examined for cell infiltration and morphology using an Olympus BX53TR microscope. All images were processed using the CellSens software (Olympus).

### 4.16. Left Ventricular Catheterization for Heart Function Test

Left ventricular catheterization was performed as described previously [55]. Briefly, for invasive hemodynamics, left ventricular catheterization was performed 3 weeks after the operation. A Millar Micro-tip 2 F pressure transducer (model SPR-838, Millar Instruments, Houston, Texas, USA) was introduced into the left ventricle via the right carotid artery under Zoletil^®^50 (0.8 mL/kg; Virbac) and Rompun (0.2 mL/kg; Bayer Korea) anesthesia. Real-time pressure-volume loops were recorded by a blinded investigator and all data were analyzed with PVAN 3.5 software (Millar).

### 4.17. Statistical Analysis

The data are expressed as the mean ± standard error of the mean of at least three independent experiments. Comparisons between two groups were performed by Student’s T-test and more than two groups were performed by one-way ANOVA analysis of variance using Bonferroni’s correction. *p* < 0.05 was considered significant.

## 5. Conclusions

In summary, the present study has provided strong evidence of M1 macrophage-mediated pyroptosis in the ischemia/reperfusion (I/R) injured hearts and identified miRNA-762 as an important regulator of IL-1β production and subsequent pyroptosis. In the clinical context, the results of this study indicate that modulating a key miRNA targeting IL-1β can be an effective means to preventing the functional demise of the heart by preventing pyroptosis following I/R injury and suppressing pyroptosis can be an effective adjuvant strategy to enhance the efficacy of stem cell-based therapeutics for diseased hearts. To fully address the unanswered issues, further studies are warranted.

## Figures and Tables

**Figure 1 ijms-22-07946-f001:**
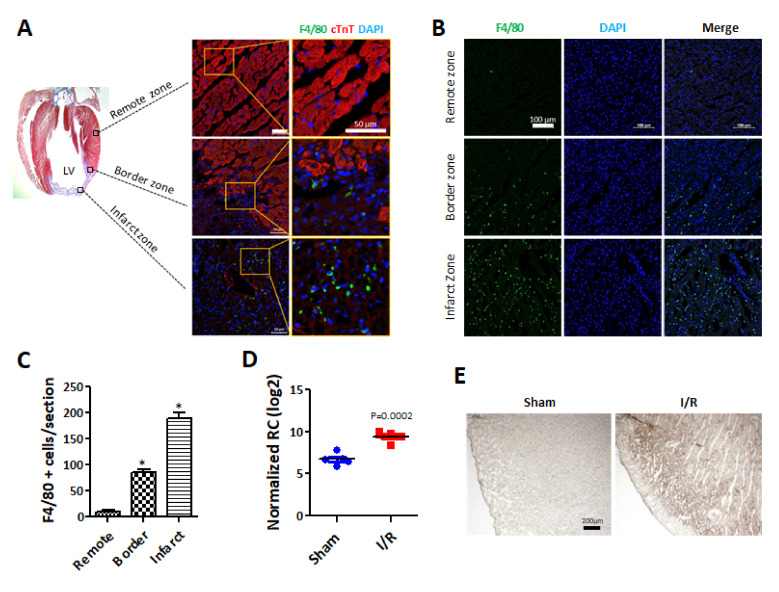
Macrophages are recruited in rat hearts with ischemia/reperfusion (I/R) injury. (**A**) Cardiac fibrosis occurred in the hearts with I/R injury after a week. Cardiac fibrosis was measured by Masson’s trichrome staining using paraffin section. Blue indicates that collagen expression is increased in the infarct zones of the hearts with I/R injury (left). Recruited macrophages are revealed by immunofluorescence staining in the hearts with I/R injury (right). Antibodies indicate the following: red: cTnT, a specific cardiomyocyte marker; green: F4/80, a specific macrophage marker; and blue: DAPI, a nuclear marker. Scale bar = 50 µm, *n* = 5 per group. (**B**) F4/80 expression was detected in the hearts one day after I/R injury. The heart sections from the acute I/R injury areas were analyzed by immunofluorescence staining with F4/80 (green: macrophages) using a frozen section. DAPI indicates nuclear staining. Scale bar = 100 μm. *n* = 3 per group (**C**) Quantitative analysis was performed for F4/80-positive cells in the heart sections. F4/80-positive cells in the border zone and infarct zone were significantly increased. * *p* < 0.05 vs remote zone (**D**) IL-1β expression levels were further analyzed by DEG analysis in sham and I/R-injured hearts. (**E**) IL-1β expression was determined by immunohistochemistry using paraffin sections one day after the induction of I/R injury in the hearts. Images were captured with 40× magnification. Scale bar = 200 μm.

**Figure 2 ijms-22-07946-f002:**
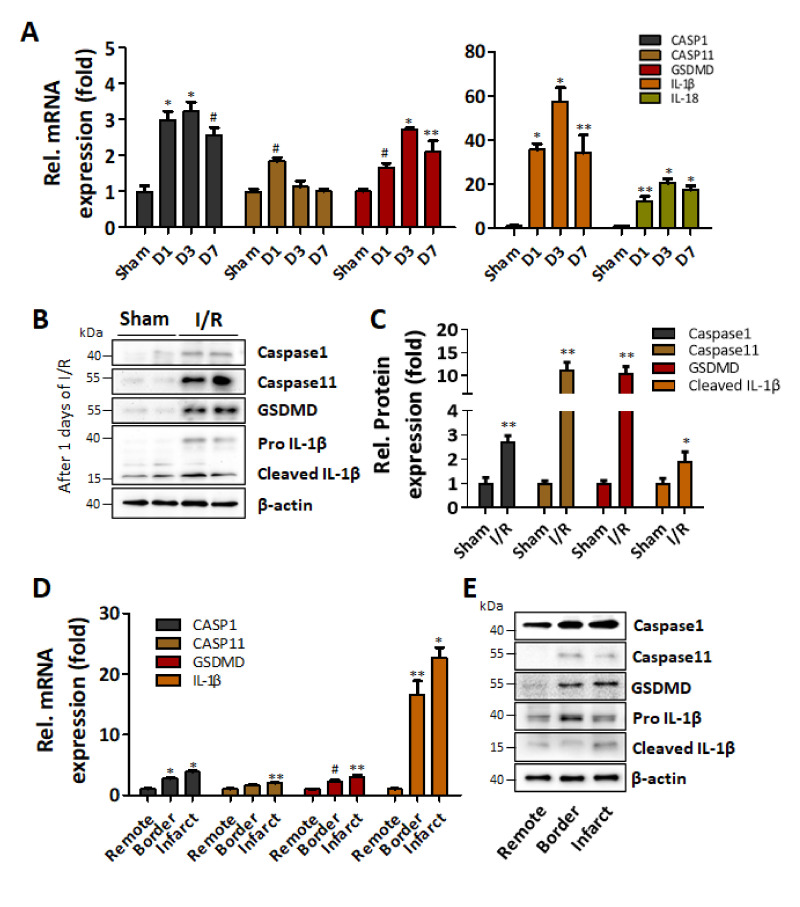
Expression of pyroptosis-related genes and proteins is increased in the heart with ischemia/reperfusion injury. (**A**) Pyroptosis-related gene expression was measured by qRT-PCR. mRNA was extracted from the infarct zones of the hearts. The expression of pyroptosis-related genes, including caspase-1, caspase-11, GSDMD, IL-1β, and IL-18, was significantly increased in the early I/R-injured hearts. CASP1 indicates caspase-1. CASP11 indicates caspase-11. GSDMD indicates gasdemin D. ^#^
*p* < 0.05 vs. normal, ** *p* < 0.01 vs. normal, * *p* < 0.001 vs. normal, (**B**) Caspase-1, caspase-11, GSDMD, and IL-1β expression levels in normal and I/R-injured hearts were detected by western blotting. (**C**) Pyroptosis-related proteins expression was normalized to β-actin. ** *p* < 0.01 vs. normal, * *p* < 0.001 vs. normal, (**D**) Caspase-1, caspase-11, GSDMD, and IL-1β mRNA expression levels by qRT-PCR were higher in the border zone and infarct zone compared than in the remote zone a day after I/R injury in the hearts. Pyroptosis-related gene expression was normalized to GAPDH. ^#^
*p* < 0.05 vs. normal, ** *p* < 0.01 vs. normal, * *p* < 0.001 vs. normal. (**E**) Caspase-1, caspase-11, GSDMD, pro-IL-1β, and cleaved IL-1β protein expression levels were higher in the border zone and infarct zone than in the remote zone a day after I/R injury in the heart.

**Figure 3 ijms-22-07946-f003:**
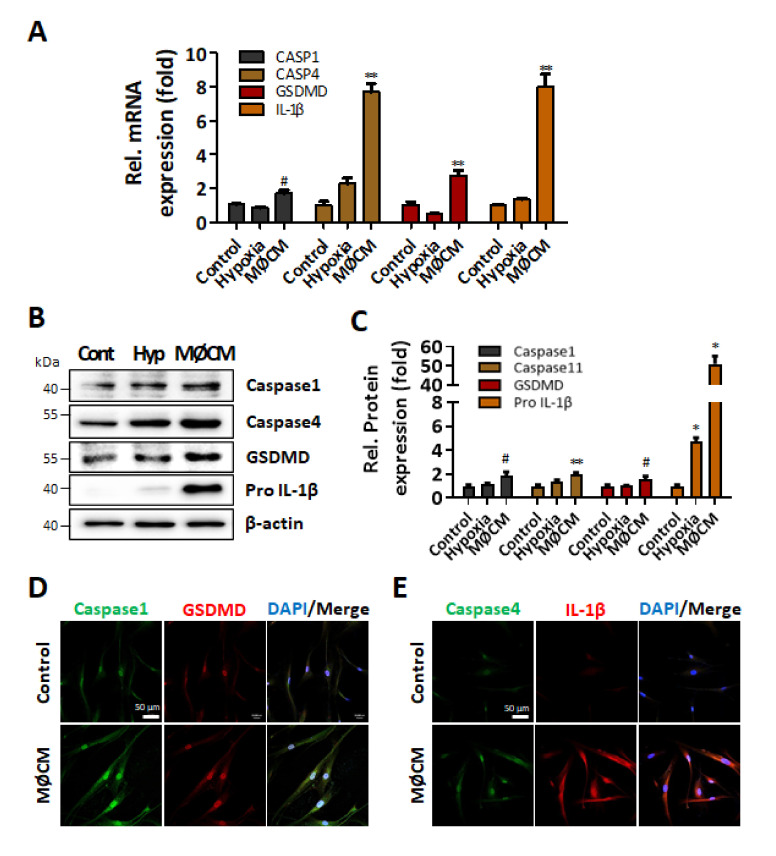
Expression of pyroptosis-related genes is increased by M1 macrophage conditioned media in ASCs. hASCs were incubated with M1-like macrophage-CM (MØCM) for 24 h. (**A**) mRNA and (**B**,**C**) protein expression levels were measured by qRT-PCR and western blotting, respectively. ^#^
*p* < 0.05 vs. Control, ** *p* < 0.01 vs. Control, * *p* < 0.001 vs. Control. (**D**) Caspase-1 and GSDMD and (**E**) Caspase-4 and IL-1β expression levels were detected by immunofluorescence. Blue indicates the nuclei. Scale bar = 50 µm.

**Figure 4 ijms-22-07946-f004:**
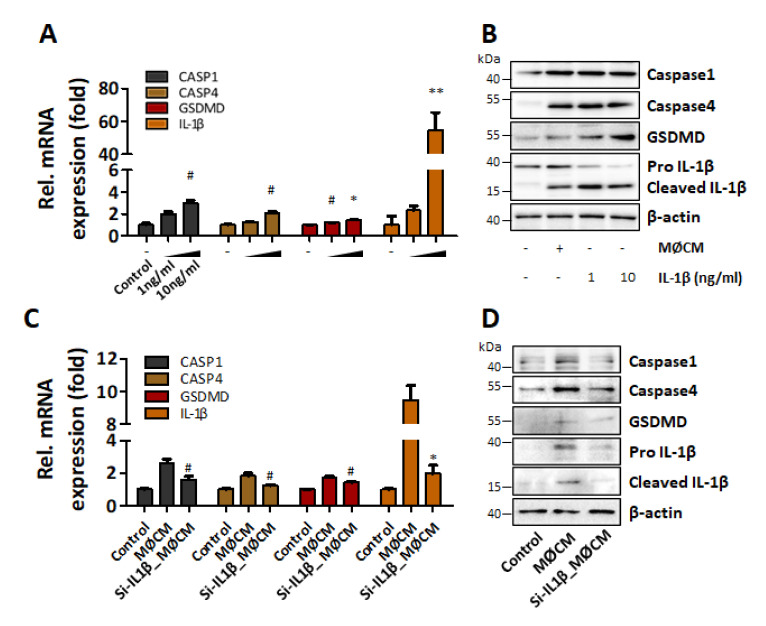
IL-1β plays an important role for pyroptosis induction in hASC. IL-1β recombinant protein at concentrations of 1 ng/mL and 10 ng/mL was used to treat hASCs for 24 h. Then, mRNA and protein levels were detected by (**A**) qRT-PCR and (**B**) western blotting. ^#^
*p* < 0.05 vs. Control, ** *p* < 0.01 vs. Control, * *p* < 0.001 vs. Control. hASCs were cultured for 24 h with IL-1β siRNA-transfected macrophage-CM (Si-IL1β_MØCM). Then, pyroptosis-related mRNA and protein levels were detected by (**C**) qRT-PCR and (**D**) western blotting, respectively. ^#^
*p* < 0.05 vs. MØCM, * *p* < 0.001 vs. MØCM.

**Figure 5 ijms-22-07946-f005:**
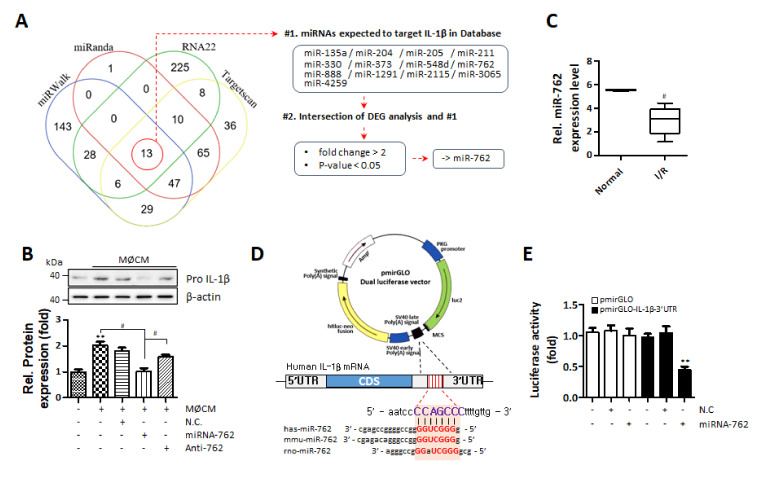
Screening miRNAs predicted to target IL-1β. miRNAs (13 miRNAs) were selected based on (**A**) 4 databases, miRanda, RNA22, miRwalk, and TargetScan (#1), and DEG analysis with criteria of *p* < 0.05 and a fold change > 2 resulted in the final selection of miRNA-762 (#2). (**B**) IL-1β protein levels in hASCs treated with miRNA-762 mimic (miRNA-762), negative control (N.C.) or anti-miRNA-762 (Anti-762) (^#^
*p* < 0.05, ** *p* < 0.01 vs. Normal). (**C**) miRNA-762 levels were quantified by DEG analysis for the normal group and the ischemic heart group. (**D**) Schematic representation of the matched miRNA-762 and 3′ UTR of IL-1β mRNA. (**E**) Luciferase activity of miRNA-762-transfected cells was measured in response to the 3′UTR of IL-1β. miRNA-762 or negative control (N.C.) was co-transfected into HeLa cells with the pmir-GLO vector or pmir-GLO containing the 3′ UTR of IL-1β as a miRNA-762 binding site. Luciferase activity was normalized to the renilla activity. (** *p* < 0.01 vs. Control).

**Figure 6 ijms-22-07946-f006:**
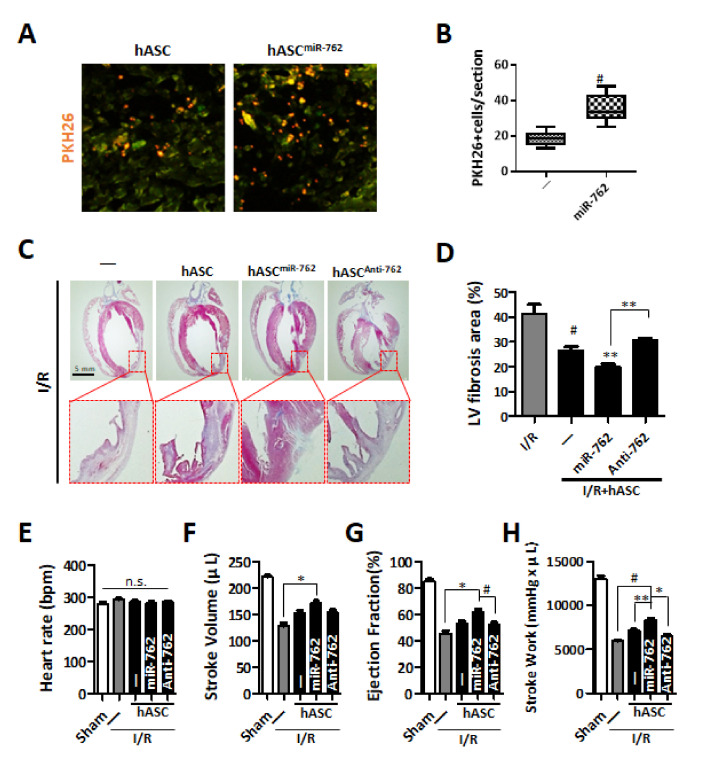
hASC^miR*−*762^ affects the reduction of fibrosis at ischemia/reperfusion injury heart. (**A**) Seven days after hASC or hASC^miR*−*762^ transplantation, PKH26-stained ASCs were detected in hearts one week after I/R-injury. Scale bar = 100 μm. (**B**) PKH26-positive cells were quantified by counting stained cells and averaged per section. ^#^
*p* < 0.05 (**C**,**D**) Fibrosis was detected by Masson’s trichrome staining in three hearts of rats per group at one week after I/R injury. ^#^
*p* < 0.05 vs. I/R group, ** *p* < 0.01 vs. I/R group. *n* = 3 per group. (**E**–**H**) Left ventricular function was measured at 3 weeks after I/R injury and hASC^miR*−*762^ transplantation. Each component of cardiac function was quantified as follows. *n* = 3 per group. (**E**) heart rate, (**F**) stroke volume, (**G**) ejection fraction, (**H**) stroke work. Each experimental group was measured five times. ^#^
*p* < 0.05, ** *p* < 0.01, * *p* < 0.001.

## Data Availability

The data that support the findings of this study are available on request from the corresponding author.

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
