# Peer review of "Suppressing Pyroptosis Augments Post-Transplant Survival of Stem Cells and Cardiac Function Following Ischemic Injury"

_ijms, 2021, doi:10.3390/ijms22157946_

Round 1
Reviewer 1 Report
Please see the attachment for details.

Author Response
17, July, 2021
Dear Editor,
Please find enclosed our revised manuscript titled “Suppressing pyroptosis augments post-transplant survival of stem cells and cardiac function following ischemic injury” which we are submitting as an original article for publication in your respected journal, International Journal of Molecular Sciences.
We authors very much appreciated the encouraging, critical and constructive comments and suggestions on this manuscript by the reviewers. The comments have been very thorough and useful in improving the manuscript. We strongly believe that the comments and suggestions have significantly increased the scientific value of revised manuscript. We are submitting the corrected manuscript with consolidated data. The manuscript has been revised as per the comments given by the reviewer, and our responses to all the comments are as follows:
Reviewer 1 Comments and Suggestions for Authors
- The citation could be at the end of the sentence or before the comma (,) not at the beginning of the sentence. Use space before the citation. Please revise the whole manuscript and correctly cite it
Response: The citation issues was been corrected as recommended.
- ‘2. Results’ section seems to be ‘Result and discussion’ as in this section authors discussed their results and related literature elaborately. Consider revision. It's better to write results and discussion separately as you wrote discussion late
Response: We removed and/or relocated a few sentences with references from “result” to either Introduction or Discussion. However, a few remained references in result are not for discussion, but for providing readers brief background to better understand the results to come.
- Figure S5.
Response: The typo (il-1β) has been corrected.
- Figure 1e _left or right panel, i.e., magnified or what is the second row
Response: The detailed description was added in Figure 1e legend. The original image set contained both 40x (upper) and 200x magnified images (bottom). However, 200x images does not give more detailed information so that the 200x image has been removed in the revised manuscript.
- Figure 5. totarget
Response: the typo has been corrected.
- Line 330-331, In this study, we presented pyroptosis as a cause of transplanted stem cells and introduced miRNA as a tool for prooving the concept. Please check. As a cause of what?
Response: the sentence in issue was ungrammatical. The sentence has been changed to “In this study, we presented pyroptosis as one of the causes of poor survival of trans-planted stem cells and utilized miRNA as a tool for proving the concept…” (page 12)
- Line 347, Cells from passages 6 to 10 were used for experiments. Please mention the reason.
Response: We mainly used hASCs from passages 6-8 for the majority of experiments and used passage 8-10 to confirm the experiments. Therefore, cell passage described in 4.1. was corrected 6 to 8.
- Line 348, Raw264.7 cells culture and Culture medium preparation. But line 349, The murine macrophage cell line RAW264.7. Write similar way across the whole manuscript.
Response: The term has been corrected to be consistent throughout the manuscript.
- Line 351, 25mM HEPES buffer. Use a space after 25.
Response : It has been corrected.
- Line 354, placed with DMEM with LPS (100ng/ml). Use a space after 100.
Response : It has been corrected.
- Line 355 IFN-γ (20ng/ml). Use a space after 20. Please use a space before each unit as used most of the time.
Response : It has been corrected as the reviewer suggested.
- Line 352, The medium was replaced with every day. Please revise the sentence and correct it.
Response : It has been corrected.
- Line 355-356, then the media is changed serum free-DMEM for 24 hours. Please revise the sentence and correct it.
Response : It has been corrected.
- Please write the detailed protocol (i.e., cell number, treatment conditions, media used, antibiotics present or absent, FBS used or not, etc.) for section ‘4.4. MiRNA transfection’, ‘4.5. Transient knockdown of IL-1β’, ‘4.10. Luciferase Assay’.
Response : Methods section has been revised to accommodate the reviewer’s suggestion.
- Line 429, intris-buffered saline with 0.05% tween 20. Use a space after ‘in’ and ‘with’.
Response : It has been corrected.
- Line 430, then incubated with the primary antibodies overnight at 4 °C. Please mention the dilution used.
Response : The dilution factor for antibodies has been added in Method section (4.9.).
- Line 450, was measured in th transfected cells 48 hours later using a luminometer. Please check.
Response : The sentence has been revised to avoid ambiguity.
- Line 458, obtained as described previously. (58). Please check.
Response : The in-text citation style has been corrected.
- Line 461, Zoletil (0.8ml/kg; Virbac, Frances) and Rompun (0.2ml/kg; Bayelkorea CO, Korea). Use a space before the unit.
Response : The space was added before units.
- Line 490, After blocking with bovine serum albumin (BSA)/PBS for 20 min. Please mention the concentration of BSA used.
Response : 1% BSA in PBS was used and this was described in 4.14.
- Line 491-495, FITC-conjugated goat anti-rabbit IgG and rhodamine-conjugated rabbit anti-mouse IgG were used as secondary antibodies. The sections with DAPI staining for nuclei were analyzed by confocal laser scanning microscope (LSM 700, Zeiss) and transferred to a computer equipped with ZEN2012 software. Please mention how DAPI staining was done. And mention how the images were analyzed for the reproducibility of the study.
Response: More detailed explanation was added in 4.14.
- Line 485, Cryosections of 5 μm of thickness and line 499, Tissue sections (5 μm thickness). Please mention how these tissue sections were prepared.
Response: More detailed explanation was added in 4.12 and 4.14.
- Line 504, please mention how H&E staining was observed, photo captured, analyzed, etc.
Response: We added more details in 4.15.
- Line 510, (0.8ml/kg; Virbac) and Rompun (0.2ml/kg; Bayelkorea CO) anesthesia. Put a space before unit.
Response : The space was added before units.
- Line 538, All authors have read and agreed to the published version of the manuscript. Published or submitted?
Response: Submitted is the correct term. It has been corrected.
- Data presented in the manuscript is enough to prove their hypothesis, but the result and discussion must be improved before publish
Response: The whole manuscript has been thoroughly revised as the reviewer suggested.
Reviewer 2 Report
Comments to the Authors:
The present manuscript is aimed to analyze the mechanism underlying acute death of transplanted stem cells in the infarcted heart. Starting from previous literature assessing a potential role of pyroptosis as an important programmed death mechanism in damaged hearts, the Authors investigated its significance in the death of transplanted stem cells after cardiac reperfusion, analyzing its activation mediated by M1 macrophages and IL1b production, and identifying miR-762 as a potential regulator of this process. Such results highlight the possible use of miR-762-implemented stem cells for cardiac transplantation to increase their post-transplantation survival and therapeutic efficacy.
Major issues:
- Quality of English language needs to be improved, there are too many grammar mistakes. The Authors should have the manuscript reviewed by someone who is fluent in English, or should use English language editing service
- In the text, the Authors reported the differential expression of three genes reported in both the inflammatory response and pyroptosis gene list, but in the table (Fig. S1B) only two of these genes are reported. What about CASP4?
- The Authors should add the description of DEG analysis and of quantitative analysis of IL1b in sham and I/R damaged hearts, which are missing in the Materials and methods section. Also the description of protein extraction from heart tissue is missing.
- The Authors should clarify some points about IL1b, because the data are not convincing. The antibody used for IL1b recognizes both the precursor (35 Kda) and the mature form (17kDa). The nomenclature in all the figures should be the same, so the Authors should decide if indicate the precursor as IL1b (as in Fig.2b) and the mature form as cleaved-IL1b or the precursor as pro-IL1b (as in Fig. 2e) and the mature form as IL1b (the last is probably the best, since the protein of interest is the mature form). The Authors should also clarify which band should be considered in the results: in Fig. 2b, cleaved-IL1b did not seem particularly increased in the I/R hearts, and there is no match with protein quantification in Fig. 2c; conversely, in Fig. 2e quantification seemed to represent the cleaved IL1b trend rather than that of pro-IL1b. Moreover, in Fig. 3 the high increase of IL1b after incubation with M1 conditioned medium is relative to the precursor form, while in figs. S3 and S4 such increase is specifically related to the cleaved form.
- Why in Fig. 2e and 4b the blot with anti-IL1b is not showed as a single panel containing both pro-IL1b and the cleaved form, since they are both recognized by the same antibody? How did the authors explain the higher increase of pro-IL1b in the border zone? Also in Fig. 3 b, both the IL1b isoforms should be shown in WB panel.
- The Authors should discuss the potential influence of using CM of murine macrophages cell line to treat human ASC cells. They should also describe the cardiomyocites and cardiac fibroblasts used in figs. S3 and S4. Are they of murine or human origin? Are they established cell lines or primary cultures? Please specify
- S2a,b: it is not clear if panel S2b represents the quantification of RT-PCR reported in Fig. S2a. If so, I think it is questionable, since in the graph ARG1 fold expression is much higher than that of IL1b, and it did not correspond to the panel a figure. The Authors should also explain how did they evaluate statistical significance of RT-PCR results. Did they reported in graph the mean ± SD of the two samples showed in panel a? ARG1 increase in LPS/IFNg-treated cells is too much evident to be considered not significant, and it should be discussed
- It is not clear why from fig 3 on the Authors evaluated caspase 4 instead of Caspase 11. Please explain
- A control ELISA assay to assess IL1b expression in si-IL1b-transfected M1 macrophages-CM should be performed to verify the transfection efficiency
- In the text, the Authors wrote: “Transplantation of hASCmiR-762 also significantly attenuated I/R-induced decline of heart function evaluated by heart rate (Figure 6E)”, but it is not correct since heart rate did not show any significant variation upon I/R injury or hASC transplantation.
Minor issues:
- In the legend of Figure 1, the description of the C panel is repeated two times, and in the E panel the scale bars values are missing
- In fig. 3c, it would be better to use only one y axis (the segmented feature can be easily applied to the principal axis, as shown in fig. 4a)
- In Fig. 3e, the legend reported Caspase-1instead of Caspase-4
- In the in vivo experiment, the Authirs should indicate the number of mice in each treatment group
Author Response
17, July, 2021
Dear Editor,
Please find enclosed our revised manuscript titled “Suppressing pyroptosis augments post-transplant survival of stem cells and cardiac function following ischemic injury” which we are submitting as an original article for publication in your respected journal, International Journal of Molecular Sciences.
We authors very much appreciated the encouraging, critical and constructive comments and suggestions on this manuscript by the reviewers. The comments have been very thorough and useful in improving the manuscript. We strongly believe that the comments and suggestions have significantly increased the scientific value of revised manuscript. We are submitting the corrected manuscript with consolidated data. The manuscript has been revised as per the comments given by the reviewer, and our responses to all the comments are as follows:
Review2 Comments and Suggestions for Authors
Comments to the Authors:
The present manuscript is aimed to analyze the mechanism underlying acute death of transplanted stem cells in the infarcted heart. Starting from previous literature assessing a potential role of pyroptosis as an important programmed death mechanism in damaged hearts, the Authors investigated its significance in the death of transplanted stem cells after cardiac reperfusion, analyzing its activation mediated by M1 macrophages and IL1b production, and identifying miR-762 as a potential regulator of this process. Such results highlight the possible use of miR-762-implemented stem cells for cardiac transplantation to increase their post-transplantation survival and therapeutic efficacy.
Major issues:
- Quality of English language needs to be improved, there are too many grammar mistakes. The Authors should have the manuscript reviewed by someone who is fluent in English, or should use English language editing service
Response: The manuscript has been thoroughly checked for grammatical errors and revised by a native speaking biologist.
- In the text, the Authors reported the differential expression of three genes reported in both the inflammatory response and pyroptosis gene list, but in the table (Fig. S1B) only two of these genes are reported. What about CASP4?
Response: We have checked the expression change of CASP4 because it is known to play an important role in pyroptosis. However, it did not show any I/R-dependent significant change (it surely is involved in both pyroptosis and inflammation, but the expression change was not significant). Therefore, the manuscript has been revised to minimize the confusion, including the number of genes reported (page 3).
- The Authors should add the description of DEG analysis and of quantitative analysis of IL-1b in sham and I/R damaged hearts, which are missing in the Materials and methods section.
Response: The description of DEG analysis and of quantitative analysis of IL-1b have been added in section 4.13 as the reviewer suggested.
3-1. Also the description of protein extraction from heart tissue is missing.
Response: The detailed additional method were added in 4.9.
- The Authors should clarify some points about IL1b, because the data are not convincing. The antibody used for IL1b recognizes both the precursor (35 Kda) and the mature form (17kDa). The nomenclature in all the figures should be the same, so the Authors should decide if indicate the precursor as IL1b (as in Fig.2b) and the mature form as cleaved-IL1b or the precursor as pro-IL1b (as in Fig. 2e) and the mature form as IL1b (the last is probably the best, since the protein of interest is the mature form).
Response: The secreted IL-1b has an actual biological activity, and it is a cleaved form of IL-1b as the reviewer mentioned. Therefore, we designated proteins as pro-IL-1b and cleaved IL-1b throughout the figures to avoid ambiguity.
4-1. The Authors should also clarify which band should be considered in the results: in Fig. 2b, cleaved-IL1b did not seem particularly increased in the I/R hearts, and there is no match with protein quantification in Fig. 2c;
Response: Quantification of IL-1b in Figure 2C was mistakenly included pro form as well. Therefore, quantification was re-done only for cleaved IL-1b.
4-2. conversely, in Fig. 2e quantification seemed to represent the cleaved IL1b trend rather than that of pro-IL1b. Moreover, in Fig. 3 the high increase of IL1b after incubation with M1 conditioned medium is relative to the precursor form, while in figs. S3 and S4 such increase is specifically related to the cleaved form.
Response: We assume the reviewer accidentally mistaken Figure 2D as the quantification data of Figure 2E. However, Figure 2D is a qRT-PCR data independent form Figure 2E.
As reviewer’s pointed out, Figure 3C shows the quantification for pro IL-1b. Unlike heart tissue or cardiomyocytes, detecting the cleaved IL-1b from hASC lysate using western blot was unsuccessful. Therefore, for hASCs, the amount of cleaved IL-1b is determined by measuring secreted IL-1b in culture media using ELISA as second best option (Figure S5).
- Why in Fig. 2e and 4b the blot with anti-IL1b is not showed as a single panel containing both pro-IL1b and the cleaved form, since they are both recognized by the same antibody?
Response : Although the same IL-1b antibody (NB600-633, Novusbio) was used to detect both pro and cleaved IL-1b as described in Materials and Methods, depending on experimental condition and cell types, the intensity between pro and cleaved bands can vary. The western imaging system we use automatically set up exposure time meaning the device automatically set up imaging condition to detect the most significant band, if there are multiple bands with different intensities. Therefore, if two adjacent bands have very different intensities, the device only detect most prominent band (for example, either only pro band is visible, or pro band is over burned if cleaved band is visible). To bypass such limitation of automated device, we usually mask one band with aluminum foil so the device only detects single band at a time. That is why we sometimes present 2 separate bands rather than two band in a single panel.
5-1. How did the authors explain the higher increase of pro-IL1b in the border zone (Fig.2e)?
Response: Our speculation on the issue is as follows; as shown in Figure 2B, I/R injury induced increase of pro IL-1b. That means either I/R condition induces IL-1b biosynthesis of host cells and/or recruited inflammatory cells. In the border zone, that is less hostile environment than the infarct zone so that not all the cells undergo cell death, surviving cells (host cells and/or recruited inflammatory cells) can still synthesize IL-1b. However, unlike the cells in border zone, the cells in the infarct zone (no matter where they are came from) are rapidly either dead or dying so that they cannot afford to synthesize new proteins. Therefore, we observe higher expression of pro-IL 1b in the border zone compare to that in the infarct zone.
5-2. Also in Fig. 3b, both the IL1b isoforms should be shown in WB panel.
Response: Unlike heart tissue or cardiomyocytes, detecting the cleaved IL-1b from hASC lysate using western blot was unsuccessful. Therefore, for hASCs, the amount of cleaved IL-1b is determined by measuring secreted IL-1b in culture media using ELISA as second best option (Figure S5).
- The Authors should discuss the potential influence of using CM of murine macrophages cell line to treat human ASC cells. They should also describe the cardiomyocytes and cardiac fibroblasts used in figs. S3 and S4. Are they of murine or human origin? Are they established cell lines or primary cultures? Please specify
Response: We used CM of mouse origin macrophage to stimulate hASCs. BLAST search indicated that mouse IL-1b is 78.65% identical to that of human at mRNA (NM_008361.4) level and 68% identical at protein level (NP_032387.1). Although our various data collectively suggest that even IL-1b of mouse origin could stimulate human ASCs so that using CM of mouse origin for this proof of concept study is acceptable, it may be necessary for further studies to use IL-1b of human origin for the study to have clinical relevance. This has been added to the discussion as one of the limitations of the present study (page 12). The cells used for figS3 and S4 are h9c2 and rat primary cultured-fibroblast, respectively. The details are described in Supplementary Materials.
- S2a,b: it is not clear if panel S2b represents the quantification of RT-PCR reported in Fig. S2a. If so, I think it is questionable, since in the graph ARG1 fold expression is much higher than that of IL1b, and it did not correspond to the panel a figure.
Response: The results from S2a was obtained by conventional RT-PCR and those of S2b was obtained by qRT-PCR, so they are independent data.
7.1. The Authors should also explain how they evaluate statistical significance of RT-PCR results. Did they reported in graph the mean ± SD of the two samples showed in panel a?
Response: S2a is data of conventional RT-PCR using duplicated samples without statistical evaluation. Again, S2b is qRT-PCR data with statistical evaluation. Sorry for the confusion we might have caused.
7.2. ARG1 increase in LPS/IFNg-treated cells is too much evident to be considered not significant, and it should be discussed
Response: Recent studies indicated that even LPS or LPS+ IFNÉ£ that are classical inducer of M1 differentiation can also increase M2 marker arg1 expression, although it is still lower than that induced by IL4 in Raw 264.7 cells [1,2]. Likewise, our data shows LPS/IFNÉ£ treatment induced ARG1 expression but it was still weaker than that by IL-4. It was about 3 fold : 8 fold for conventional RT-PCR and 8 fold: 16 fold for qRT-PCR.
- It is not clear why from fig 3 on the Authors evaluated caspase 4 instead of Caspase 11. Please explain
Response: We described the relation between caspase 4 and caspase 11 in Introduction. To explain further, although caspase-4 and caspase-5 are human orthologs of rodent caspase-11 [3,4], many of the conclusions regarding their immunological functions have been based on research on murine caspase-11. Despite the similarities between caspase-11 and caspase-5, most publications consider caspase-4 to be the functional homolog of caspase-11 [5]. The antibody we used in this experiments can detect caspase 11 of rat and mouse both.
- A control ELISA assay to assess IL1b expression in si-IL1b-transfected M1 macrophages-CM should be performed to verify the transfection efficiency
Response: We perform the ELISA for si-IL1b in Raw264.7 and added result as Figure S6B.
- In the text, the Authors wrote: “Transplantation of hASCmiR-762 also significantly attenuated I/R-induced decline of heart function evaluated by heart rate (Figure 6E)”, but it is not correct since heart rate did not show any significant variation upon I/R injury or hASC transplantation.
Response: As, reviewer’s suggestion, we revised the sentence in issue (2.1.7).
Minor issues:
- In the legend of Figure 1, the description of the C panel is repeated two times, and in the E panel the scale bars values are missing
Response: The duplication was removed from Figure 1 legend.
- In fig. 3c, it would be better to use only one y axis (the segmented feature can be easily applied to the principal axis, as shown in fig. 4a)
Response: The graph was fixed as reviewer’s suggestion.
- In Fig. 3e, the legend reported Caspase-1 instead of Caspase-4
Response: The error was corrected.
- In the in vivo experiment, the Authors should indicate the number of mice in each treatment group
Response: The number of rat was added in Figure 6 legend.
References
- Imrie, H.; Williams, D.J.L. Stimulation of bovine monocyte-derived macrophages with lipopolysaccharide, interferon-, Interleukin-4 or Interleukin-13 does not induce detectable changes in nitric oxide or arginase activity. BMC Vet Res 2019, 15, 45, doi:10.1186/s12917-019-1785-0.
- Menzies, F.M.; Henriquez, F.L.; Alexander, J.; Roberts, C.W. Sequential expression of macrophage anti-microbial/inflammatory and wound healing markers following innate, alternative and classical activation. Clin Exp Immunol 2010, 160, 369-379, doi:10.1111/j.1365-2249.2009.04086.x.
- Shi, J.; Zhao, Y.; Wang, K.; Shi, X.; Wang, Y.; Huang, H.; Zhuang, Y.; Cai, T.; Wang, F.; Shao, F. Cleavage of GSDMD by inflammatory caspases determines pyroptotic cell death. Nature 2015, 526, 660-665, doi:10.1038/nature15514.
- Zhao, Y.; Shi, J.; Shao, F. Inflammatory Caspases: Activation and Cleavage of Gasdermin-D In Vitro and During Pyroptosis. Methods Mol Biol 2018, 1714, 131-148, doi:10.1007/978-1-4939-7519-8_9.
- Casson, C.N.; Yu, J.; Reyes, V.M.; Taschuk, F.O.; Yadav, A.; Copenhaver, A.M.; Nguyen, H.T.; Collman, R.G.; Shin, S. Human caspase-4 mediates noncanonical inflammasome activation against gram-negative bacterial pathogens. Proc Natl Acad Sci U S A 2015, 112, 6688-6693, doi:10.1073/pnas.1421699112.
Reviewer 3 Report
Lee et. al. insisted that pyroptosis contributes to stem cell death as its cause when stem cell is transplanted into an ischemic injury cardiac mouse model. Especially, it did that pyroptosis of stem cells is done by control of miR-762 and IL-1. The explanation of the mechanism of stem cell death with pyroptosis is a subject of research which can be helpful for stem cell researchers. However, there is a lack of evidence to support the hypotheses insisted by the authors generally and the experimental techniques are thought to be required to be supplemented.
The results provided by the author are as follows: I/R injury has found that M1 macrophage was recruited by injury. M1 macrophage increased the transcription level of IL-1 and caused pyroptosis. (Figure 1-2). However, MOCM or IL-1 was treated in ASC, expression of IL-1 rapidly increased in ASC and its pyroptosis did (Figure 3-4). This concluded that ASC transplanted by IL-1 secreted by M1 rapidly increases expression of IL-1 and does injury of host heart by ASC transplanted. However, it is thought that these results contradict the results of Figure 6 that the fibrosis area is improved when ASC is solely transplanted. This explanation is required.
There is little effect of antisense-miR-762 used in the paper. This result of the experiment is very unfavorable for proving the authors’ hypotheses. Antisense results are required to prove the hypothesis that miR-762 can directly control IL-1 and pyroptosis. And there should be scramble control in all the studies on microRNA. And an experiment of luciferase was conducted to show that miR-762 directly control the post-transcriptional level of IL-1. For this experiment, results of research including antisense-miR-762 and IL-1-mutation sequence should be added.
Author Response
17, July, 2021
Dear Editor,
Please find enclosed our revised manuscript titled “Suppressing pyroptosis augments post-transplant survival of stem cells and cardiac function following ischemic injury” which we are submitting as an original article for publication in your respected journal, International Journal of Molecular Sciences.
We authors very much appreciated the encouraging, critical and constructive comments and suggestions on this manuscript by the reviewers. The comments have been very thorough and useful in improving the manuscript. We strongly believe that the comments and suggestions have significantly increased the scientific value of revised manuscript. We are submitting the corrected manuscript with consolidated data. The manuscript has been revised as per the comments given by the reviewer, and our responses to all the comments are as follows:
Reviwer3 Comments and Suggestions for Authors
Lee et. al. insisted that pyroptosis contributes to stem cell death as its cause when stem cell is transplanted into an ischemic injury cardiac mouse model. Especially, it did that pyroptosis of stem cells is done by control of miR-762 and IL-1. The explanation of the mechanism of stem cell death with pyroptosis is a subject of research which can be helpful for stem cell researchers. However, there is a lack of evidence to support the hypotheses insisted by the authors generally and the experimental techniques are thought to be required to be supplemented. The results provided by the author are as follows: I/R injury has found that M1 macrophage was recruited by injury.
- M1 macrophage increased the transcription level of IL-1 and caused pyroptosis. (Figure 1-2). However, MOCM or IL-1 was treated in ASC, expression of IL-1 rapidly increased in ASC and its pyroptosis did (Figure 3-4).This concluded that ASC transplanted by IL-1 secreted by M1 rapidly increases expression of IL-1 and does injury of host heart by ASC transplanted. However, it is thought that these results contradict the results of Figure 6 that the fibrosis area is improved when ASC is solely transplanted. This explanation is required.
Response: As reviewer’s comment, it is reasonable to postulate that transplanted ASCs can secret IL-1b by hostile microenvironment, and thus have harmful effect on host heart. However, it has been known that stem cells secret various factors having protective and therapeutic potential [1,2], and the secretion of IL-1b from the transplanted ASCs are initiated and amplified only when they are stimulated by hostile microenvironment such as IL-1b secreted by M1 macrophage according to our data. That means that the ASCs transplanted into the border zone where relatively lower hostile stimuli exist compared to the infarct zone still have beneficial effects over harmful effects induced by IL-1b, although the IL-1b induced harmful effects possibly negate the beneficial effects of the transplanted ASCs in the infarct zone. Therefore, transplantation of ASC only can improve cardiac fibrosis and cardiac function as we observed in the present study and many other previous studies. Furthermore, miR-762 enriched ASCs further improved cardiac fibrosis and cardiac function supporting such speculation.
- There is little effect of antisense-miR-762 used in the paper. This result of the experiment is very unfavorable for proving the authors’ hypotheses. Antisense results are required to prove the hypothesis that miR-762 can directly control IL-1 and pyroptosis.
Response: In the early phase of the present study, our primary interest was more on whether miR-762 suppresses IL-1b expression rather than on whether miR-762 directly targets IL-1b. Furthermore, although it was not for all experiments, we have utilized antisense (Figure 5B). Nevertheless, as the reviewer pointed out, using antisense for majority of the experiments could have been strengthen our manuscript for sure. Therefore, although is it limited, we have conducted an additional experiment using antisense to examine pyroptosis-associated protein levels such as caspase1, caspase4, or GSDMD. The additional result was included as Figure S7.
- And there should be scramble control in all the studies on microRNA.
Response: We used negative control which was purchased with miRNA-762 mimic from Genolution (http://www.genolution1.com). According to manufacturer, negative control is a small RNA which does not show homologous effect in mouse, rat, and human. Although the underlying mechanism may differ, this still could be compatible to scramble control in functional aspects. Furthermore, due to the current pandemic situation, the supply of synthesized nucleotides is lagging in Korea so that we most likely would not complete all the experiments using antisense within the given revision period even if we tried. We authors know this cannot be very strong reasons not to accommodate the reviewer’s suggestion, but just hope that the reviewer know it was not out of disrespect of the reviewer’s comment.
- And an experiment of luciferase was conducted to show that miR-762 directly control the post-transcriptional level of IL-1. For this experiment, results of research including antisense-miR-762 and IL-1-mutation sequence should be added.
Response: As we stated above, in the early phase of the present study, our primary interest was more on whether miR-762 suppresses IL-1b expression rather than on whether miR-762 directly targets IL-1b. Again, as the reviewer pointed out, using antisense for luciferase assay also could have been strengthen our manuscript. However, due to the current pandemic situation, the supply of synthesized nucleotides is lagging in Korea so that we most likely would not complete all the experiments using antisense within the given revision period even if we tried. We authors just hope that the reviewer know it was not out of disrespect of the reviewer’s comment. We also added not using antisense for luciferase assay as one of the limitations of the present study.
- Lin, L.; Du, L. The role of secreted factors in stem cells-mediated immune regulation. Cell Immunol 2018, 326, 24-32, doi:10.1016/j.cellimm.2017.07.010.
- Fan, X.L.; Zhang, Y.; Li, X.; Fu, Q.L. Mechanisms underlying the protective effects of mesenchymal stem cell-based therapy. Cell Mol Life Sci 2020, 77, 2771-2794, doi:10.1007/s00018-020-03454-6.
Round 2
Reviewer 2 Report
The authors satisfactorily addressed my previous concerns. However, I have a remaining concern that need to be addressed to improve the manuscript:
- The Authors added the description of the IL-1β Enzyme-linked Immunosorbent Assay (ELISA) in the supplementary methods, but they mentioned Raw264.7 cells and the use of a mouse IL-1 beta ELISA kit (Invitrogen, while the experiments reported in the manuscript have been performed on human ASCs. Please clarify this point.
Author Response
- The Authors added the description of the IL-1β Enzyme-linked Immunosorbent Assay (ELISA) in the supplementary methods, but they mentioned Raw264.7 cells and the use of a mouse IL-1 beta ELISA kit (Invitrogen, while the experiments reported in the manuscript have been performed on human ASCs. Please clarify this point.
Response: Sorry for the confusion, the information on the human IL-1b ELISA had been accidentally omitted from the 1st revised manuscript. We have added the information to the Supplementary methods.
Reviewer 3 Report
I am satisfied with your revising the manuscript according to the suggested changes and found it much clear. I don't have any more comments on this manuscript. I think it is suitable for publication.
Author Response
I am satisfied with your revising the manuscript according to the suggested changes and found it much clear. I don't have any more comments on this manuscript. I think it is suitable for publication.
-We authors very much appreciated the encouraging, critical and constructive comments and suggestions on this manuscript by the reviewer. The comments have been very thorough and useful in improving the manuscript.